# A Continuous Mapping For Augmentation Design

**Keyu Tian**[*]
Software College
Beihang University
tiankeyu.00@gmail.com

**Chen Lin**[*]
University of Oxford
chen.lin@eng.ox.ac.uk

**Ser-Nam Lim**
Facebook AI

**Wanli Ouyang**
University of Sydney

**Puneet K. Dokania**
University of Oxford & Five AI Ltd.

**Philip H.S. Torr**
University of Oxford
philip.torr@eng.ox.ac.uk

## Abstract

Automated data augmentation (ADA) techniques have played an important role in boosting the performance of deep models. Such techniques mostly aim to optimize a parameterized distribution over a *discrete* augmentation space. Thus, are restricted by the discretization of the search space which normally is handcrafted. To overcome the limitations, we take the first step to constructing a continuous mapping from $\mathbb{R}^d$ to image transformations (an augmentation space). Using this mapping, we take a novel approach where **1)** we pose the ADA as a continuous optimization problem over the parameters of the augmentation distribution; and **2)** use Stochastic Gradient Langevin Dynamics to learn and sample augmentations. This allows us to potentially explore the space of infinitely many possible augmentations, which otherwise was not possible due to the discretization of the space. This view of ADA is radically different from the standard discretization based view of ADA , and it opens avenues for utilizing the vast efficient gradient-based algorithms available for continuous optimization problems. Results over multiple benchmarks demonstrate the efficiency improvement of this work compared with previous methods.

## 1 Introduction

Data augmentation [22] is one of the advanced techniques or recipes that proliferate the success of deep learning. The crux behind data augmentation is to make a model invariant towards various input transformations that we expect to observe during inference, thus, improves the generalization of the model. Since it is not clear which transformations would benefit most, manual design is difficult. Thus, there has been a growing interest in *automatically learning* such augmentations, a new thriving subarea of data augmentation known as the Automated Data Augmentation (ADA) [5, 16, 21, 20, 19, 28].

ADA is generally formulated as a bilevel optimization problem over a *discrete* search space, and aims to find an optimal augmentation policy. For example, the search space used in AutoAugment [5], the very first promising work in this topic, contains several augmentation candidates, *e.g.*, rotation/blurring/brightening with different discrete magnitudes, and the augmentation policy is defined as a parameterized discrete distribution over this space. Such approach has been shown to provide

---
[*]Authors contributed equally.

35th Conference on Neural Information Processing Systems (NeurIPS 2021).

far more superior performances over several datasets and tasks compared against the traditional handcrafted augmentation counterparts. This, however, comes at the expense of a *high computational budget*, taking far more time than model training, resulting in an untenable training procedure in the presence of real-world large-scale datasets. While several approaches have been proposed to address the efficiency issue [20, 16, 21, 19], the compromise in performance due to such efficiency measures was nontrivial; some of the current state-of-art remains computationally expensive [28]. The *contradiction* between performance and efficiency has raised a key challenge in this field.

Another major limitation of such approaches is due to the discretized search space which limits the diversity of the learnable policy. In fact, the discretization results in augmentations that do not always represent the variations in the real world sufficiently. In the real world, the data is collected with a great range of freedom such as varying camera viewpoint and lighting conditions. However, the discrete augmentation would only be able to produce transformed data with specific types and magnitudes. Furthermore, quantization of augmentation search space introduces extra manual design, which is contrary to the aim of ADA.

In this paper, we resolve the aforementioned issues of discretization by constructing a continuous and differentiable mapping $\phi$ which maps an augmentation vector $\boldsymbol{\alpha} \in \mathbb{R}^d$ to an image transformation function $t_{\boldsymbol{\alpha}} \in \mathcal{T}$ that transforms an image $I$ to $t_{\boldsymbol{\alpha}}(I)$. Specifically, we group basic augmentations into three categories: **Color Adjustment** that adjusts the color values, **Image Filtering** like blurring or sharpening, and **Image Warping** which represents the geometric transformation. We then split the augmentation vector $\boldsymbol{\alpha}$ to three sub-vectors ($\boldsymbol{\alpha}_{\mathrm{CA}}$, $\boldsymbol{\alpha}_{\mathrm{IF}}$ and $\boldsymbol{\alpha}_{\mathrm{IW}}$), each of which would map to a specific transformation ($t_{\boldsymbol{\alpha}_{\mathrm{CA}}}$, $t_{\boldsymbol{\alpha}_{\mathrm{IF}}}$ or $t_{\boldsymbol{\alpha}_{\mathrm{IW}}}$). We finish the construction by regard $t_{\boldsymbol{\alpha}}$ as the composite transformation, *i.e.*, $t_{\boldsymbol{\alpha}} = t_{\boldsymbol{\alpha}_{\mathrm{IW}}} \circ t_{\boldsymbol{\alpha}_{\mathrm{IF}}} \circ t_{\boldsymbol{\alpha}_{\mathrm{CA}}}$, as visualized in Fig. 1. Based on this construction, we pose ADA as learning an augmentation distribution in the continuous augmentation space. This gives us the freedom to search over the entire continuous space in order to obtain the optimal augmentations. For optimization and inference, we use the well-known Stochastic Gradient Langevin Dynamics (SGLD) and propose a *pseudo one-step* approximation that allows us to efficiently learn the underlying augmentation distribution and sample augmentations from it. On a wide range of experiments involving state-of-the-art architectures on multiple image classification benchmarks, we show that our approach not only provides results at par with existing ADA methods, it also is orders of magnitude faster than them.

Our main contributions can be summarized as:

1. We propose a continuous augmentation mapping and first pose ADA as a continuous optimization problem, thus, avoiding the pitfalls of the discrete augmentation space.

2. We develop a gradient-based approach to learn the augmentation distribution in the continuous space that allows for highly efficient sampling of augmentations, offering a favorable speed-performance trade-off.

## 2   Related Work

**Recent automated augmentation methods**   have been proposed that optimize data augmentation policies on different datasets and tasks [5, 16, 21, 20, 19, 28]. AutoAugment [5] provides a discrete augmentation search space and an RNN controller to parameterize the augmentation distribution. Reinforcement learning is then employed to search for the augmentation strategy. Subsequent work focuses more on searching efficiently in the same space. To this end, OHL [21] proposed an online reinforcement learning pipeline to reduce the computation complexity. PBA [16] proposed a population based search strategy to obtain the augmentation distribution with fitness selection also in an online manner. Fast AutoAugment [20] proposed a different objective function that applies data augmentation at test time. AWS [28] investigates the effectiveness of a weight-sharing paradigm, yielding superior performance on various datasets. The most related work, DADA [19] take a continuous relaxation to the discrete augmentation distribution by Gumbel-softmax to obtain better approximation. Our work eliminated such approximation tricks when computing the gradient thanks to the continuous nature of the proposed search space.

**Markov Chain Monte Carlo (MCMC)**   is a popular sampling method for approximating the posterior distribution in large-scale Bayesian modeling. MCMC draws samples from high-dimensional

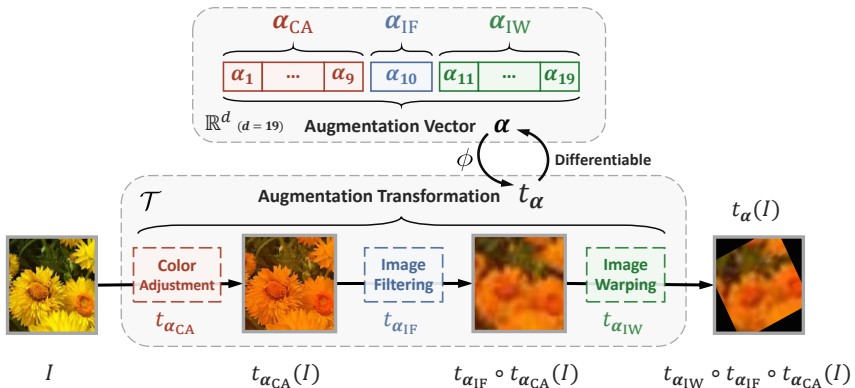

Figure 1: **Schematic representation of the continuous image augmentation mapping** $\phi : \boldsymbol{\alpha} \mapsto t_{\boldsymbol{\alpha}}$. $t_{\boldsymbol{\alpha}}$ is the composite transformation of color adjustment $t_{\boldsymbol{\alpha}_{\mathrm{CA}}}$ (red in the figure), image filtering $t_{\boldsymbol{\alpha}_{\mathrm{IF}}}$ (blue), and image warping $t_{\boldsymbol{\alpha}_{\mathrm{IW}}}$ (green). Notably, the augmentation process is fully differentiable w.r.t. $\boldsymbol{\alpha}$ (explained in Sec. 3.2).

probability distributions in the construction of a Markov chain. A broad category of recent MCMC methods, namely Stochastic Gradient MCMC (SG-MCMC) [3, 1, 9, 30], comes from the traditional stochastic gradient optimization and thus inherits the efficiency of Stochastic Gradient Descent (SGD). To cast a stochastic optimization process to a sampling procedure, artificial noise needs to be added to the gradients. A typical example of gradient MCMC methods is Stochastic Gradient Langevine Dynamics (SGLD) [30], which injects Gaussian noise into the parameter gradients, pushing the optimization trajectory to asymptotically converge to the actual posterior distribution. In practice, some of these algorithms are also used in combination with several heuristic approximations, such as constant step sizes or diagonal approximations to the Hessian for the sake of simplicity or efficiency [25]. In Sec. 4.2 of this paper, we also use the simple SGLD with constant noise to optimize our augmentation distribution.

## 3  A Continuous and Differentiable Mapping

### 3.1  Preliminaries

Let the mapping $\phi : \boldsymbol{\alpha} \mapsto t_{\boldsymbol{\alpha}}$ map a $d$-dimensional vector $\boldsymbol{\alpha} \in \mathbb{R}^d$ to an image augmentation transformation $t_{\boldsymbol{\alpha}} \in \mathcal{T}$, where the augmentation $t_{\boldsymbol{\alpha}}(\cdot)$ maps an image $I$ to its transformed one $t_{\boldsymbol{\alpha}}(I)$. For convenience, we introduce the notation $t_{\boldsymbol{\alpha}}$ to denote the transformation function in $\mathcal{T}$ for which $t_{\boldsymbol{\alpha}} = \phi(\boldsymbol{\alpha})$. In other words, $\boldsymbol{\alpha}$ could be regarded as the parameter of the transformation function $t_{\boldsymbol{\alpha}}$, and $\phi$ is the realization function.

### 3.2  Constructing The Mapping

We start by developing a taxonomy of image augmentation transformations. Popular basic augmentations can be grouped into three categories: **Color Adjustment** that adjusts the color values pixel by pixel, **Image Filtering** like blurring or sharpening the whole picture, and **Image Warping** which applies geometric transformations to the image. Now we discuss how to construct a continuous and differential mapping for each augmentation categories respectively:

**Color Adjustment** is defined as a transformation in the spatial domain that is *equally* applied to each pixel. Let $n$ be the image size. For every location $(x, y)$ with pixel vector $I_{xy} = [\mathrm{h}_{xy} \ \mathrm{s}_{xy} \ \mathrm{v}_{xy}]^{\mathsf{T}}$, it would be adjusted to $I'_{xy} = \left[\mathrm{h}'_{xy} \ \mathrm{s}'_{xy} \ \mathrm{v}'_{xy}\right]^{\mathsf{T}}$. The augmentation can be described as:

$$I' = t_{\boldsymbol{\alpha}_{\mathrm{CA}}}(I), \ \text{ where } I'_{xy} = \begin{bmatrix} \mathrm{h}'_{xy} \\ \mathrm{s}'_{xy} \\ \mathrm{v}'_{xy} \end{bmatrix} = \begin{bmatrix} \alpha_{\mathrm{h}} & + & (1 + \beta_{\mathrm{h}})(\mathrm{h}_{xy})^{\gamma_{\mathrm{h}}} \\ \alpha_{\mathrm{s}} & + & (1 + \beta_{\mathrm{s}})(\mathrm{s}_{xy})^{\gamma_{\mathrm{s}}} \\ \alpha_{\mathrm{v}} & + & (1 + \beta_{\mathrm{v}})(\mathrm{v}_{xy})^{\gamma_{\mathrm{v}}} \end{bmatrix} \ (\forall \, x, y \text{ that } 1 \le x, y \le n),$$

$$(1)$$

where the nine-dimensional vector $\boldsymbol{\alpha}_{\text{CA}} = [\alpha_{\text{h}} \ \beta_{\text{h}} \ \gamma_{\text{h}} \ \alpha_{\text{s}} \ \beta_{\text{s}} \ \gamma_{\text{s}} \ \alpha_{\text{v}} \ \beta_{\text{v}} \ \gamma_{\text{v}}]^{\mathsf{T}}$ maps to a Color Adjustment augmentation $t_{\boldsymbol{\alpha}_{\text{CA}}}(\cdot)$. The first three values control the hue, the next three would change the saturation, and the last three control the brightness. This representation is motivated by the Power law (Gamma) transformation function [24]. As shown in Equ. (1), this augmentation family is differential w.r.t. all $\alpha, \beta, \gamma$ since it can be expressed as power functions of them and both RGB-to-HSV and HSV-to-RGB conversions are known to be continuous and differentiable [26, 29].

**Image Filtering** is a typical image enhancement in the frequency domain. Here we consider using filters to blur or sharpen images. Let a single value $s$ denote the degree of sharpening or smoothing. Let $G_k$ be a sharpness filter obtained by zero-mean normalizing a $k \times k$ discrete Gaussian filter, and $C_k$ be a $k \times k$ identity filter which has an unique non-zero value 1.0 at its center. The sharpening or smoothing process that transforms an image $I$ to $I'$ is:

$$I' = t_{\boldsymbol{\alpha}_{\text{IF}}}(I) = I * (s \cdot G_k + C_k), \tag{2}$$

where the asterisk $*$ means convolution and $t_{\boldsymbol{\alpha}_{\text{IF}}}(\cdot)$ is the Image Filtering function. If $s$ is greater than zero, it represents sharpening. Otherwise, it is equivalent to blurring. Thus, a one-dimensional vector $[\boldsymbol{\alpha}_{\text{IF}}]^{\mathsf{T}}$ could map to an Image Filtering transformation.

Image filtering is differentiable w.r.t $s$ by nature. However, the computation grows fast with respect to the filter size $k$. When $k$ is close to the image size $n$, a better implementation for computing the convolution would be applying discrete Fast Fourier Transform (FFT) and multiplying the frequencies, which achieve a complexity of $\mathcal{O}(n^2 \log n)$. In this work, we take direct convolution since we pick $k \ll n$ for simplicity.

**Image Warping** refers to a geometric distortion on an image where pixels are moved to other locations without changing their pixel values. Perspective transformation (also known as *projective transformation*) is a popular warping method, which can be uniquely represented by a $3 \times 3$ matrix $\mathbf{H}$ called *homography* [2]. Each point in the original image $(x, y)$ would move to $(x', y')$ with the perspective transformation. This Image Warping can be described as:

$$
\begin{aligned}
&I' = t_{\boldsymbol{\alpha}_{\text{IW}}}(I) \\
&\iff I'_{x'y'} = I_{xy}, \quad \text{where} \quad
\begin{cases}
x' = \dfrac{\mathbf{H}_{11}x + \mathbf{H}_{12}y + \mathbf{H}_{13}}{\mathbf{H}_{31}x + \mathbf{H}_{32}y + \mathbf{H}_{33}}, \\[2mm]
y' = \dfrac{\mathbf{H}_{21}x + \mathbf{H}_{22}y + \mathbf{H}_{23}}{\mathbf{H}_{31}x + \mathbf{H}_{32}y + \mathbf{H}_{33}},
\end{cases}
\quad (\forall \, x, y \text{ that } 1 \le x, y \le n),
\end{aligned}
\tag{3}
$$

where $t_{\boldsymbol{\alpha}_{\text{IW}}}(\cdot)$ denotes the Image Warping augmentation via perspective transformation, and $\boldsymbol{\alpha}_{\text{IW}} = [\mathbf{H}_{11} \ \mathbf{H}_{12} \ \mathbf{H}_{13} \ \mathbf{H}_{21} \ \mathbf{H}_{22} \ \mathbf{H}_{23} \ \mathbf{H}_{31} \ \mathbf{H}_{32} \ \mathbf{H}_{33}]^{\mathsf{T}}$ could represent an Image Warping augmentation. Rotations, translations, shearing, scaling, as well as more single geometric transformations and any combinations of them are all special cases of perspective transformation. Perspective transformation is originally non-differentiable to each element in the homography $\mathbf{H}$ since index $x, y$ take discrete values. However, we can leverage interpolation between pixels and inverse warping to make the transformation differentiable [17, 11, 27].

**In summary,** an augmentation vector $\boldsymbol{\alpha} = [\boldsymbol{\alpha}_{\text{CA}}^{\mathsf{T}} \ \boldsymbol{\alpha}_{\text{IF}}^{\mathsf{T}} \ \boldsymbol{\alpha}_{\text{IW}}^{\mathsf{T}}]^{\mathsf{T}}$ lies in continuous real space with $d = 9 + 1 + 9 = 19$ dimensions could map to a specific augmentation transformation $t_{\boldsymbol{\alpha}} = t_{\boldsymbol{\alpha}_{\text{IW}}} \circ t_{\boldsymbol{\alpha}_{\text{IF}}} \circ t_{\boldsymbol{\alpha}_{\text{CA}}}$. Although composite order matters, for simplicity, leave the matter to future work. One can easily verify that many commonly used augmentation transforms [5, 6, 20, 16, 21, 19, 28], such as brightening, blurring, and rotation, are all discrete samples in the augmentation function space $\mathcal{T}$ of our mapping. Last but not least, practical constraint functions (`tanh`, `exp`) and clamp function are used before and after applying augmentations respectively, to ensure all augmentations are valid and all augmented pixels are in $[0, 1]$. The overview of this space is illustrated in Fig. 1. More details on the augmentation process of this figure are provided in Supplementary A.

### 3.3 An Example Random Augmentation Policy

An example random augmentation policy in our *continuous* augmentation space is provided here to evaluate the space by comparing this policy with other policies in *discrete* space. This random policy is based on a multivariate Gaussian distribution $g(\boldsymbol{\alpha})$ over $\boldsymbol{\alpha}$ with zero means and diagonal covariance. The according augmentation distribution is denoted as $p_{\text{rand}}(t_{\boldsymbol{\alpha}})$, where $t_{\boldsymbol{\alpha}} = \phi(\boldsymbol{\alpha})$ and $\boldsymbol{\alpha} \sim g(\boldsymbol{\alpha})$. This policy is one of the simplest continuous random policy one could define.

Table 1: **Comparison between our example random policies in continuous augmentation space and others based on discrete space.** Top-1 errors of Wide-ResNet-40-2 on CIFAR-10 are reported. Lower is better.

| Method | Example Policy (random, std=0.5) | Example Policy (random, std=1.0) | AutoAugment (random, our impl.) | AutoAugment (searched, from [5]) | DADA (searched, from [19]) |
|---|---|---|---|---|---|
| Error | **3.25** | 3.34 | 4.08 | 3.7 | 3.5 |

In Tab. 1 we compare the example random policy with both random and optimized policies in *discrete* augmentation space to show the advance of *continuity*. Note that DADA [19] basically follows the same discrete space of AutoAugment's [5]. Two example policies with different standard deviations (0.5 and 1.0) are evaluated for better comparison.

## 4 Automatic Augmentation Design using $\phi$

### 4.1 ADA Problem formulation

Let $t(\cdot)$ denotes the augmentation function. ADA aims to find the optimal distribution of augmentations $q(t)$, which improves the generalization of the model. Let $\mathcal{D}_t = \{(\mathbf{x}_i, \mathbf{y}_i)\}_{i=1}^{N_t}$ and $\mathcal{D}_v = \{(\tilde{\mathbf{x}}_i, \tilde{\mathbf{y}}_i)\}_{i=1}^{N_v}$ be the training and validation datasets, respectively. Training a deep network $f_\theta$ parameterized by $\theta$ is the process of minimizing a loss $\mathcal{L}(\cdot, \cdot)$, which is cross-entropy in this work. We use $p_\theta(\mathbf{y}|\mathbf{x}_i)$ to refer to the probability distribution over labels by $f_\theta$ for $\mathbf{x}_i$, *i.e.*, the output of $f_\theta$ followed by a `softmax` function. In general, ADA aims at solving the following bilevel optimization:

$$\underset{q(t)}{\arg\min} \sum_{(\tilde{\mathbf{x}}_i, \tilde{\mathbf{y}}_i) \in \mathcal{D}_v} \mathcal{L}(f_{\theta^*}(\tilde{\mathbf{x}}_i), \tilde{\mathbf{y}}_i), \tag{4}$$

$$\text{s.t. } \theta^* = \underset{\theta}{\arg\min} \sum_{\substack{(\mathbf{x}_i, \mathbf{y}_i) \in \mathcal{D}_t, \\ t \sim q(t)}} \mathcal{L}(f_\theta(t(\mathbf{x}_i)), \mathbf{y}_i), \tag{5}$$

where $q(t)$ is the distribution of augmentations, $t(\mathbf{x}_i)$ denotes the augmented data of $\mathbf{x}_i$. The goal is to minimize the outer-level validation loss of a network $f_{\theta^*}$ in (4) that is trained by minimizing the inner-level training loss using *augmented* training data in (5).

Notably, by writing the cross-entropy explicitly, we will have:

$$\mathcal{L}(f_\theta(x), y) = -\log(p_\theta(y|x)), \tag{6}$$

for any model parameter $\theta$, and any labed data pair $(x, y)$.

### 4.2 Gradient-Based ADA in Continuous Augmentation Space via SGLD

**ADA in continuous space.** Previous work mostly defines $q(t)$ as a parameterized categorical distribution or a discrete uniform distribution. In this work, given the proposed continuous mapping $\phi : \mathbb{R}^d \to \mathcal{T}$ where $\mathcal{T}$ is the set of augmentations, we are now able to relate the augmentation distribution $q(t)$ for $t \in \mathcal{T}$ to a continuous distribution $p(\boldsymbol{\alpha})$ for $\boldsymbol{\alpha} \in \mathbb{R}^d$ by $t = t_{\boldsymbol{\alpha}} = \phi(\boldsymbol{\alpha})$. Thus we can reform the bilevel optimization (4)(5) by substituting the target distribution $q(t)$ with $p(\boldsymbol{\alpha})$ and apply the realized transformation $t_{\boldsymbol{\alpha}}$ to training data $\mathbf{x}_i$ then get (7)(8):

$$\arg\min_{p(\boldsymbol{\alpha})} \sum_{(\tilde{\mathbf{x}}_i, \tilde{\mathbf{y}}_i) \in \mathcal{D}_v} \mathcal{L}(f_{\theta^*}(\tilde{\mathbf{x}}_i), \tilde{\mathbf{y}}_i), \tag{7}$$

$$\text{s.t. } \theta^* = \arg\min_{\theta} \sum_{\substack{(\mathbf{x}_i, \mathbf{y}_i) \in \mathcal{D}_t, \\ \boldsymbol{\alpha} \sim p(\boldsymbol{\alpha})}} \mathcal{L}(f_\theta(t_{\boldsymbol{\alpha}}(\mathbf{x}_i)), \mathbf{y}_i). \tag{8}$$

**One-step simplified inner-level.** Notably, directly solving the bilevel optimization in (7)(8) would be prohibitive since it requires hundreds or thousands of full model training and validation. Thus, we refer to the successful augmentation-wise weight-sharing [28] and one-step meta learning [4, 10, 23] techniques to make this problem more tractable.

Specifically, we begin with an augmentation-wise shared parameter $\theta_s$ on $\mathcal{D}_t$ using the random augmentation policy $p_{\text{rand}}(\boldsymbol{\alpha})$ introduced in Sec. 3.3. We replace the inner optimization (8) with a *pseudo one-step* update: a single gradient step for a batch of training samples $\mathcal{B}_t = \{(\mathbf{x}_i, \mathbf{y}_i)\}_{i=1}^{M}$ augmented by a given $t_{\boldsymbol{\alpha}}$ is taken to get $\bar{\theta}_s$ that approximates the $\theta^*$, where $M$ is the batch size. The inner optimization in (8) now becomes:

$$\theta^* \approx \bar{\theta}_s = \theta_s - \lambda \sum_{(\mathbf{x}_i, \mathbf{y}_i) \in \mathcal{B}_t} \nabla_{\theta_s} \mathcal{L}(f_{\theta_s}(t_{\boldsymbol{\alpha}}(\mathbf{x}_i)), \mathbf{y}_i). \tag{9}$$

**Bayesian formulation for ADA.** In this setup, ADA boils down to a continuous distribution optimization problem (7)(9). In order to apply gradient-based methods, we drive a new Bayesian formulation for distribution optimization in ADA . Let $p_{\text{prior}}(\boldsymbol{\alpha})$ be the prior distribution and $p(\boldsymbol{\alpha}|\mathcal{D}_v)$ be the posterior distribution. The optimization in (7) for obtaining the target distribution $p(\boldsymbol{\alpha})$ is modeled in our Bayesian formulation by inferring the posterior distribution $p(\boldsymbol{\alpha}|\mathcal{D}_v)$ as follows:

$$p(\boldsymbol{\alpha}|\mathcal{D}_v) \propto p_{\text{prior}}(\boldsymbol{\alpha}) \prod_{i=1}^{N_v} p\Big((\tilde{\mathbf{x}}_i, \tilde{\mathbf{y}}_i)|\boldsymbol{\alpha}\Big), \tag{10}$$

where $\prod_{i=1}^{N_v} p\Big((\tilde{\mathbf{x}}_i, \tilde{\mathbf{y}}_i)|\boldsymbol{\alpha}\Big)$ is the data likelihood of the validation set given a fixed augmentation parameter $\boldsymbol{\alpha}$. After obtaining $\bar{\theta}_s$ using Equ. (9) for a given $\boldsymbol{\alpha}$, this likelihood now can be computed efficiently as $\prod_{i=1}^{N_v} p_{\bar{\theta}_s}(\tilde{\mathbf{y}}_i|\tilde{\mathbf{x}}_i)$. Note that the inner-level optimization is efficient as we always start with the same $\theta_s$ and then take a single gradient step over the given batch under the augmentation $t_{\boldsymbol{\alpha}} = \phi(\boldsymbol{\alpha})$ to obtain $\bar{\theta}_s$. Assuming uniform prior $p_{\text{prior}}(\boldsymbol{\alpha})$, the posterior in (10) can be computed as:

$$p(\boldsymbol{\alpha}|\mathcal{D}_v) \propto \prod_{i=1}^{N_v} p_{\bar{\theta}_s}(\tilde{\mathbf{y}}_i|\tilde{\mathbf{x}}_i). \tag{11}$$

**Use gradients to search for augmentation policy.** With the Bayesian formulation and the proposed differentiable mapping $\phi$, Gradient-MCMC methods like Stochastic Gradient Langevin Dynamics (SGLD) is now applicable to find the target distribution by sampling. Recall Equ. (6), the $j$-th iteration of SGLD sampling on the validation batch $\mathcal{B}_v = \{(\tilde{\mathbf{x}}_i, \tilde{\mathbf{y}}_i)\}_{i=1}^{M}$ can be described as:

$$\boldsymbol{\alpha}^{(j+1)} = \boldsymbol{\alpha}^{(j)} + \eta + \frac{\epsilon}{2} \frac{M}{N_v} \sum_{(\tilde{\mathbf{x}}_i, \tilde{\mathbf{y}}_i) \in \mathcal{B}_v} \nabla_{\boldsymbol{\alpha}^{(j)}} \log p_{\bar{\theta}_s}(\tilde{\mathbf{y}}_i|\tilde{\mathbf{x}}_i) \tag{12}$$

$$= \boldsymbol{\alpha}^{(j)} + \eta + \frac{\epsilon M \lambda}{2N_v} \sum_{(\tilde{\mathbf{x}}_i, \tilde{\mathbf{y}}_i) \in \mathcal{B}_v} \sum_{(\mathbf{x}_k, \mathbf{y}_k) \in \mathcal{B}_t} \frac{\partial \mathcal{L}(f_{\bar{\theta}_s}(\tilde{\mathbf{x}}_i), \tilde{\mathbf{y}}_i)}{\partial \bar{\theta}_s} \frac{\partial^2 \mathcal{L}(f_{\theta_s}(t_{\boldsymbol{\alpha}^{(j)}}(\mathbf{x}_k), \mathbf{y}_k))}{\partial \theta_s \partial \boldsymbol{\alpha}^{(j)}}, \tag{13}$$

where $\epsilon$ is the step size of SGLD and $\eta \sim \mathcal{N}(0, \epsilon)$ is the Gaussian noise.

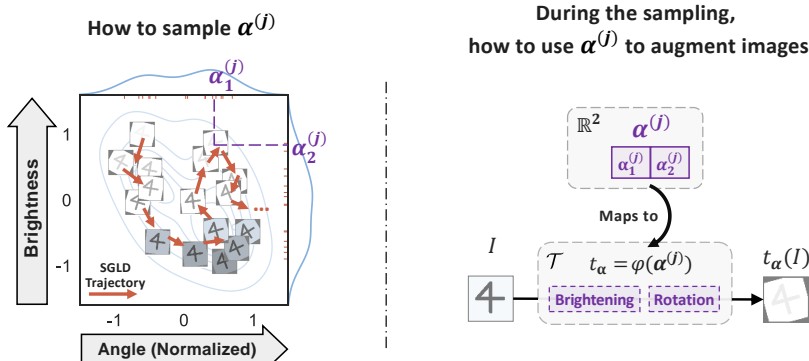

Figure 2: **Augmentation search phase of a simplified MCMC-Aug.** Suppose $\boldsymbol{\alpha} \in \mathbb{R}^2$ ($d = 2$) and the two augmentations are *brightening* and *rotation* for simplicity. **Left**: we collect the samples of $\boldsymbol{\alpha}^{(j)}$ (where $j = 1, 2, \cdots$) along the SGLD trajectory. **Right**: how to perform the augmentation.

In practice, one can use finite difference [13, 23] to compute the second-order derivative in (13) more efficiently. By iterating Equ. (13), samples of $\boldsymbol{\alpha}$ can be collected and SGLD would asymptotically converge to the posterior distribution $p(\boldsymbol{\alpha}|\mathcal{D}_v)$ [32], which is the resulted augmentation policy.

**The overview.**   The whole search pipeline of our method (MCMC-Aug) is summarized in Alg. 1. The augmentation search phase of a simplified (suppose $d = 2$) MCMC-Aug is also visualized in Fig. 2. The samples $\{\boldsymbol{\alpha}^{(1)}, \boldsymbol{\alpha}^{(2)}, \cdots, \boldsymbol{\alpha}^{(N_j)}\}$ collected by SGLD form the resulted augmentation policy. When using the policy to train a new model, one would pick randomly from these samples to augment each training image. In Sec. 5 and Supplementary E. we will evaluate and visualize the searched policy in detail.

---

**Algorithm 1** A Reference Method for ADA : MCMC-Aug

---

**Inputs:**  Training set $\mathcal{D}_t$, validation set $\mathcal{D}_v$, pretrained weights $\theta_s$, samples of the augmentation vector $\mathcal{S}$;
**Hyperparameters:**  Number of updates $N_j$, mini batch size $M$, model learning rate $\lambda$, SGLD step size $\epsilon$;
**Initiate:**  $\mathcal{S} := \varnothing$; $\boldsymbol{\alpha}^{(0)}$;
**for** $j = 0, 1, 2, \cdots, N_j - 1$ **do**
    Sample bastches $\mathcal{B}_t$ and $\mathcal{B}_v$;
    Perform pseudo one-step update in Equ. (9) to obtain $\bar{\theta}_s$;
    Perform SGLD update in Equ. (13) to get $\boldsymbol{\alpha}^{(j+1)}$;
    $\mathcal{S} := \mathcal{S} \cup \{\boldsymbol{\alpha}^{(j+1)}\}$;
**end for**
**return** $\mathcal{S}$;

---

# 5   Experiments

## 5.1   Implementation Details

We evaluate our method on CIFAR-10/100 [18] and ImageNet [7]. For each datasets, a validation set is split from the training set and the testing set is only used for evaluating the final performance (not involved in the search phase). For ImageNet, a reduced ImageNet-120 is used for fair comparison [5, 20, 16, 19]. All batch sizes for augmentation search mentioned in this work are set to 256 for CIFARs and 1024 for ImageNet as standard. More details are included in Supplementary. B.

**Search.**   Following prior work, we search policy for two base models, which are Wide-ResNet-40-2 [31] and ResNet-50 [15] on CIFARs and ImageNet respectively. Before we start, an instance of each base model is pretrained (CIFARs/ImageNet) using the random policy introduced in Sec. 3.3 to obtain the shared weights $\theta_s$ in Equ. (9). This procedure lasts 200 epochs for CIFARs and 120 epochs for ImageNet. Then we run our MCMC-Aug with 200 epochs, the same as [16] over the training set. Step size and noise scale of SGLD is set according to [25]. The experiments reported have a fixed 0.4

step size, and a noise rate of $2 \times 10^{-5}$. Further analysis on the choice of hyperparameters could be found in Supplementary D., where we provide a sensitivity analysis as well as a guideline for tuning SGLD. On CIFAR-100 and ImageNet, the same SGLD hyperparameters are adopted without any modification.

**Evaluation.** We evaluate MCMC-Aug by measuring the performance of a reinitialized model trained with augmentation from the SGLD samples $\{t_{\boldsymbol{\alpha}^{(1)}}, t_{\boldsymbol{\alpha}^{(2)}}, \cdots, t_{\boldsymbol{\alpha}^{(N_j)}}\}$. Theoretically, SGLD will be in its Langevin dynamics phase and sampling approximately from the posterior distribution when its sample threshold statistic is much smaller than 1.0 [30]. However, in practice, we found that collecting samples of $\boldsymbol{\alpha}$ from the later half of the SGLD trajectory is sufficient to obtain good performance. We run SGLD only for base models, and the collected samples could be used to retrain other models to assess the transferability. Thus Wide-ResNet-{40-2, 28-10} [31], Shake-Shake (26 $2 \times 32$d) [12], Pyramid-Net+ShakeDrop [14] are retrained on CIFARs, and ResNet-{50, 200} on ImageNet. Each of experiments

Table 2: **Test error rates on ImageNet.** Top-1 / Top-5 errors are reported. Lower is better.

| Method | ResNet-50 | ResNet-200 |
|---|---|---|
| Baseline | 23.7 / 6.9 | 21.5 / 5.8 |
| AA | 22.4 / 6.2 | 20.0 / 5.0 |
| FAA | 22.4 / 6.3 | 19.4 / 4.7 |
| RA | 22.4 / 6.2 | – |
| OHL | 21.07 / 5.68 | – |
| AWS | **20.61 / 5.49** | **18.64** / 4.67 |
| DADA | 22.5 / 6.5 | – |
| MCMC | 20.98 / 5.60 | 18.87 / **4.62** |

is repeated four times and the mean value is reported, see Supplementary B. for the error bar. Other hyperparameters not mentioned here are directly imported from [28], provided in Supplementary C.

## 5.2 Comparison with State-of-the-Arts

**Summary.** We conducted comprehensive comparisons with state-of-the-art augmentation methods that includes Cutout [8], AutoAugment (AA) [5], Fast AutoAugment (FAA) [20], Rand Augment (RA) [6], Population Based Augmentation (PBA) [16], and OHL-AutoAug (OHL) [21]. Furthermore, we compare MCMC-Aug with two of the latest ADA methods: **1)** AWS-AutoAug (AWS) [28], which is currently the best-performing ADA method reported; **2)** Differentiable Automatic Data Augmentation (DADA) [19], which is currently the fastest method that applies a continual relaxation to the discrete search space using Gumbel-softmax.

**Performance comparison.** The performance of MCMC-Aug and other methods are compared in Tab. 2 and Tab. 3. The performance of the other methods is directly obtained from the corresponding papers. We use dashes wherever the results are not reported. Following prior work [5, 19], "Baseline" in the tables refers to a model trained using only default pre-processing like normalization and basic augmentations such as random cropping and random horizontal flipping. The same pre-processing and basic augmentations are also uniformly applied to all the methods. Cutout is used in all the ADA methods on CIFAR datasets.

Table 3: **Top-1 test errors on two CIFAR datasets.** Lower is better. "MCMC" represents MCMC-Aug.

| Model | Baseline | Cutout | AA | FAA | RA | DADA | OHL | PBA | AWS | MCMC |
|---|---|---|---|---|---|---|---|---|---|---|
| *CIFAR-10* | | | | | | | | | | |
| WRN-40-2 | 5.3 | 4.1 | 3.7 | 3.7 | – | 3.5 | – | – | – | **2.96** |
| WRN-28-10 | 3.87 | 3.08 | 2.6 | 2.7 | 2.7 | 2.6 | 2.61 | 2.58 | **1.95** | 1.97 |
| Shake-Shake | 2.86 | 2.56 | 2.0 | 1.9 | 2.0 | 2.0 | – | 2.03 | 1.65 | **1.53** |
| PyramidNet | 2.67 | 2.31 | 1.5 | 1.7 | 1.5 | 1.7 | – | 1.46 | 1.31 | **1.29** |
| *CIFAR-100* | | | | | | | | | | |
| WRN-40-2 | 26.0 | 25.2 | 20.7 | 20.6 | – | 20.9 | – | – | – | **19.07** |
| WRN-28-10 | 18.8 | 18.4 | 17.1 | 17.3 | 16.7 | 17.5 | – | 16.73 | **15.28** | 15.64 |
| Shake-Shake | 17.1 | 16.0 | 14.3 | 14.6 | – | 15.3 | – | 15.31 | 14.07 | **13.98** |
| PyramidNet | 13.99 | 12.19 | 10.7 | 11.7 | – | 11.2 | – | 10.94 | **10.40** | 10.48 |

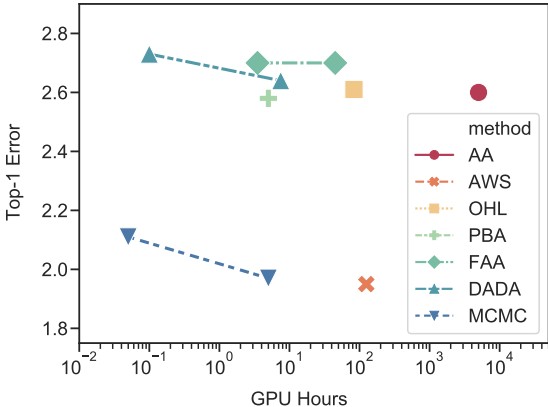

Figure 3: **Efficiency vs. performance among ADA methods.** Some points are for full CIFAR-10 and the others are for reduced CIFAR-10.

Tab. 2 and Tab. 3 show that MCMC-Aug significantly reduces the error rates over the baseline and Cutout on a variety of network architectures, demonstrating the effectiveness and transferability of the resulted policies. Compared with other ADA methods, MCMC-Aug performs on par with AWS while outperforming the others, even though we did not conduct any comprehensive parameter tuning. Moreover, in Sec. 5.3, we show that increasing the number of searching epochs (proportional to $N_j$ in Alg. 1) brings extra performance gains. Since we had only trained for 200 epochs for efficiency, we believe there is still room for improvements.

**Efficiency comparison.** We compare the computational cost of MCMC-Aug with others to check if MCMC-Aug inherits the efficiency of gradient-based optimization. Results on both the *reduced* and the *full* CIFAR-10 dataset are reported to ensure fairness, since previous work had reported efficiency of their methods on *one* or *both* of the datasets. We used the same reduced CIFAR-10 subset in [5, 20, 16, 19], which consists of 4,000 images from the training set. We kept the same hyperparameters except for a smaller number of epochs in searching (five) and linearly scaled batch size and learning rate (four times). The results are summarized in Tab. 4 and visualized in Fig. 3.

As shown in Tab. 4, OHL, FAA and DADA are much faster than AA, but at the expense of accuracy. AWS achieves the best performance, but is much more computationally expensive. Like PBA, MCMC-Aug strikes a good balance between accuracy and efficiency, but yields much better accuracy. We also note that there is a noticeable gap between MCMC-Aug's performance on the full and reduced dataset, showing the potential for further performance gains when more computational budget becomes available. Overall, our results show that MCMC-Aug is the best in class when considering the need to balance performance and efficiency.

Table 4: **Comparison of the efficiency (GPU hours).** Top-1 test error changes of Wide-ResNet-28-10 relative to AA are listed. −: not reported. †: approximated, from [19]. ↑: error increased. ↓: error decreased.

| Method | AA | AWS | PBA | OHL | FAA | DADA | MCMC |
|---|---|---|---|---|---|---|---|
| GPU Device | P100 | V100 | TitanXP | V100[†] | V100 | TitanXP | V100 |
| *Reduced CIFAR-10* | | | | | | | |
| GPU Hours | 5000 | − | 5 | − | 3.5 | 0.1 | 0.05 |
| Error Changes (normalized) | 0.0% | − | ↓0.8% | − | ↑3.9% | ↑5.0% | ↓16.15% |
| *Full CIFAR-10* | | | | | | | |
| GPU Hours | − | 125 | − | 83.3[†] | 45[†] | 7.5[†] | 5 |
| Error Changes (normalized) | − | ↓25.0% | − | ↑0.4% | ↑3.9% | ↑1.5% | ↓24.2% |

## 5.3 Ablation Study

**Space decomposition.** To show the necessity of three base catogories of augmentation mapping, we remove two categories of transformation in turns, after which we look at the change in test accuracy.

As reported in Tab. 5, the performance of Wide-ResNet-40-2 on CIFAR-10 would dramatically decrease, which suggests that each of them is indispensable.

**SGLD early-stopping.**   To investigate the optimization behavior of the SGLD, we stop collecting the SGLD samples at 25%, 50%, and 75% of the total epochs and evaluate the corresponding performance respectively. As listed in Tab. 6, we see that the errors of the distributions arising from early-stopping gradually increase. It shows the performance gain along with the SGLD procedue. This also suggests that increasing the number of searching epochs of our approach might bring extra performance gains.

Table 5: **Ablation study** on the importance of different augmentation categories.

| Remaining Category | Error |
|---|---|
| All | 2.96 |
| Color Adjustment | $3.32_{(\uparrow 0.36)}$ |
| Image Warping | $3.45_{(\uparrow 0.49)}$ |
| Image Filtering | $3.61_{(\uparrow 0.65)}$ |

Table 6: **Ablation study** on the effectiveness of SGLD.

| SGLD Early-Stopping at | Error |
|---|---|
| 100% (original) | 2.96 |
| 75% | $3.06_{(\uparrow 0.10)}$ |
| 50% | $3.15_{(\uparrow 0.19)}$ |
| 25% | $3.20_{(\uparrow 0.24)}$ |

For the sensitivity of hyperparameters and more details on the searched distribution, please see Supplementary D. and E.

## 6   Limitation

**Non-differentiable augmentations**   that do not belong to any categories of the three are a clear limitation. However, an interpolation between the original image and its augmented counterpart could be made to bypass this issue. Suppose we have an image $I$ and a *non-differentiable* transformation $t$, we can introduce a *continuous* parameter $\lambda$ and create a new augmentation $t'$ via $t'(I) = \lambda\, t(I) + (1 - \lambda)\, I$. $t'$ is a surrogate augmentation which ensures the differentiability.

**The presence of a clean validation set**   is vital to our MCMC-aug as well as other automated data augmentation algorithms. This is because the loss or accuracy on validation set is a common criteria for searching the augmentation policy. We expect to see a better approach to reduce the dependency on validation set in the future. It is a valuable direction and ought to be further explored.

## 7   Conclusion

A continuous mapping to model a continuous space of augmentation is proposed. The continuous mapping can be applied to many augmentation design tasks (*e.g.*, instance discrimination contrastive learning, image alignment, and domain adaptation). We apply this mapping to pose ADA as a continuous augmentation search problem. Furthermore, we setup a Bayesian formulation for ADA to enable the application of efficient gradient-based MCMC, presenting a new way to conduct ADA search via posterior sampling. Comprehensive comparison and studies verify its efficiency and effectiveness. Both source codes and checkpoints will be released to the public, and future work may concern more image modalities like infrared/X-rays/ultrasound imaging, more diverse augmentations, or wider applicants, *e.g.*, in other computer vision areas or the times-series processing.

## 8   Funding Disclosure and Acknowledgement

This work is supported by the EPSRC grant: Turing AI Fellowship: EP/W002981/1, EPSRC/MURI grant EP/N019474/1. Chen Lin is supported by Facebook AI. We would also like to thank the Royal Academy of Engineering and FiveAI.

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
