# A  Augmentation Details

This section provides more details on the augmentation process of Fig. 1. In Fig. 1, an example augmentation vector $\boldsymbol{\alpha} = [\boldsymbol{\alpha}_{\mathrm{CA}}^{\mathsf{T}}\ \boldsymbol{\alpha}_{\mathrm{IF}}^{\mathsf{T}}\ \boldsymbol{\alpha}_{\mathrm{IW}}^{\mathsf{T}}]^{\mathsf{T}}$ is given and the mapped augmentation transformation $t_{\boldsymbol{\alpha}}$ transforms the image $I$ to $t_{\boldsymbol{\alpha}}(I)$. The detailed setting is:

$$
\begin{aligned}
\boldsymbol{\alpha}_{\mathrm{CA}} &= [\alpha_{\mathrm{h}}\ \beta_{\mathrm{h}}\ \gamma_{\mathrm{h}}\ \alpha_{\mathrm{s}}\ \beta_{\mathrm{s}}\ \gamma_{\mathrm{s}}\ \alpha_{\mathrm{v}}\ \beta_{\mathrm{v}}\ \gamma_{\mathrm{v}}]^{\mathsf{T}} \\
&= [0\ {-0.4}\ 0\ 0\ 0\ 0.6\ 0\ 0\ 0.6]^{\mathsf{T}},
\end{aligned}
\tag{1}
$$

$$
\boldsymbol{\alpha}_{\mathrm{IF}} = [s]^{\mathsf{T}} = [-1.5]^{\mathsf{T}},
\tag{2}
$$

$$
\boldsymbol{\alpha}_{\mathrm{IW}} = [\mathbf{H}_{11}\ \mathbf{H}_{12}\ \mathbf{H}_{13}\ \mathbf{H}_{21}\ \mathbf{H}_{22}\ \mathbf{H}_{23}\ \mathbf{H}_{31}\ \mathbf{H}_{32}\ \mathbf{H}_{33}]^{\mathsf{T}},
$$
$$
\text{where } \mathbf{H} = \mathbf{R} \times \mathbf{S} = \begin{bmatrix} \cos(\pi/6) & -\sin(\pi/6) & 0 \\ \sin(\pi/6) & \cos(\pi/6) & 0 \\ 0 & 0 & 1 \end{bmatrix} \times \begin{bmatrix} 0.8 & 0 & 0 \\ 0 & 0.8 & 0 \\ 0 & 0 & 1 \end{bmatrix}
\tag{3}
$$
$$
= \begin{bmatrix} 0.69 & -0.40 & 0.00 \\ 0.40 & 0.69 & 0.00 \\ 0.00 & 0.00 & 1.00 \end{bmatrix}.
$$

For Color Adjustment (CA), $\beta_{\mathrm{h}}$ is set to $-0.4$ so that all hue values are twisted, making the whole picture look more "red"; the brightness and saturation are also enhanced with $\gamma_{\mathrm{s}} = 0.6$ and $\gamma_{\mathrm{v}} = 0.6$.

For Image Filtering (IF), $s$ equals to $-1.5$, so the image is blurred by convolving with $\mathbf{K} = -1.5\,G3 + C3$, where

$$
\mathbf{K} = -1.5 \begin{bmatrix} -0.042 & -0.083 & -0.042 \\ -0.083 & 0.5 & -0.083 \\ -0.042 & -0.083 & -0.042 \end{bmatrix} + \begin{bmatrix} 0 & 0 & 0 \\ 0 & 1 & 0 \\ 0 & 0 & 0 \end{bmatrix} = \begin{bmatrix} 0.063 & 0.125 & 0.063 \\ 0.125 & 0.25 & 0.125 \\ 0.063 & 0.125 & 0.063 \end{bmatrix}.
\tag{4}
$$

Finally, the adjusted and blurred image is zoomed out and rotated via the Image Warping (IW) transformation to get the resulted picture $t_{\boldsymbol{\alpha}}(I)$.

# B  More Details on Datasets and Results

## B.1  Datasets

**CIFAR.**  Both CIFAR-10 and CIFAR-100 [4] have 50,000 training images and 10,000 testing images in total, all of which have a resolution of $32 \times 32$. On both datasets, we run MCMC-Aug on the full training sets, and each of them is partitioned into a training subset with 40,000 samples and a validation set with 10,000 samples. Testing sets are not involved in our augmentation search process.

**ImageNet.**  ImageNet [2] is a challenging large scale dataset, containing about 1.28 million training images and 50,000 testing images from 1,000 classes. Following [1, 6, 3, 5], 120 classes are selected and the corresponding images form the reduced "ImageNet-120". A subset with 6,000 images is left out as the validation set. The testing set is not used.

## B.2  Results with Error Bars

We repeat each experiment on CIFAR-10 or CIFAR-100 for four times with different random seeds, and report the results with error bars in Tab. A.

Table A: **Test errors with error bars on two CIFAR datasets.** Mean values and standard deviations are reported.

| Dataset | WRN-40-2 | WRN-28-10 | Shake-Shake | PyramidNet |
|---|---|---|---|---|
| CIFAR-10 | $2.96 \pm 0.07$ | $1.97 \pm 0.07$ | $1.53 \pm 0.05$ | $1.29 \pm 0.04$ |
| CIFAR-100 | $19.07 \pm 0.21$ | $15.64 \pm 0.14$ | $13.98 \pm 0.15$ | $10.48 \pm 0.18$ |

## C   Detailed Hyperparameters

The hyperparameters for re-training used in this paper are listed in Tab. B. Basically, we use the same as [7]'s. For those not reported in [7], we refer to [1].

Table B: **Hyperparameters used in re-training models.** Cosine annealing is adopted for all learning rates.

| Dataset | Model | Batch Size | Learning Rate | Epochs |
|---|---|---|---|---|
| CIFARs | WRN-40-2 | 256 | 0.4 | 300 |
| | WRN-28-10 | 256 | 0.4 | 300 |
| | Shake-Shake | 512 | 0.01 | 1800 |
| | PyramidNet | 1024 | 0.8 | 1800 |
| ImageNet | ResNet-50 | 4096 | 1.6 | 300 |
| | ResNet-200 | 4096 | 1.6 | 300 |

## D   Sensitivity of Hyperparameters

It is widely observed in prior research that the practical application of SGLD requires careful hyperparameter selection. Here, we present a sensitivity analysis for two key hyperparameters, the step size and the noise rate in SGLD to serve as a guideline when selecting hyperparameters for MCMC-Aug. The step size of SGLD is set to 0.4, and we use a constant noise rate of $2 \times 10^{-5}$. It appears from the results presented in Fig. A that the impact on performance caused by different setting are less than 0.3% error rate. We share the same step size and noise scale across all the experiments without noticing any significant degradation.

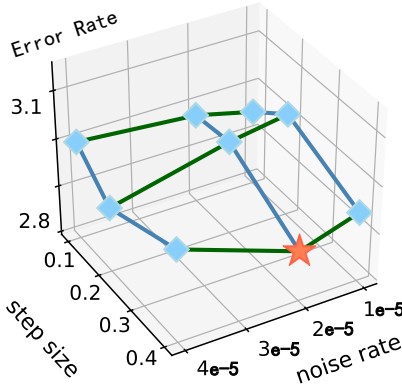

Figure A: **An illustration of how the two key hyperparameters influence the final performance.** Experiments are run on CIFAR-10 with Wide-ResNet-40-2.

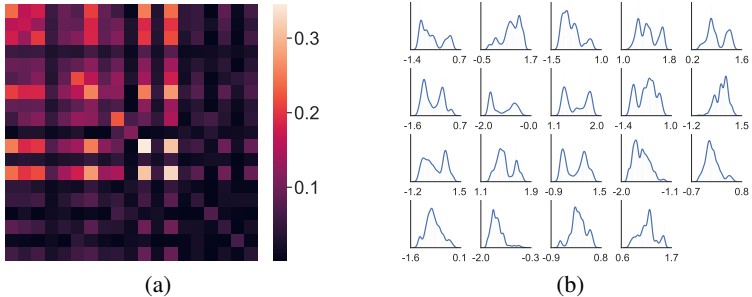

|     |     |
| --- | --- |
| (a) | (b) |

Figure B: **Visualizing some details of the posterior distribution estimated on CIFAR-10.** (a) The covariance matrix ($19 \times 19$) of the posterior. (b) 19 marginal distributions of the posterior.

# E    Details on the Searched Distribution

In this section, we seek to visualize some details of the posterior distribution estimated via MCMC-Aug on CIFAR-10. Figure 2(a) shows that the covariance matrix of the posterior is clearly not diagonal, which indicates that many augmentation components are closely related to each other. For instance, one can observe a high covariance in the upper-left $3 \times 3$ sub-matrix, which indicates that the three types of color adjustment transformations are highly related to each other.

We then try to visualize the approximated posterior distribution in Fig. 2(b). It can steer away from augmentations that destroy the information content in the image, *e.g.*, sets the total brightness to 0. Since our augmentation random variable lies in 19-dimensional space, we draw 19 marginal distributions of the original joint distribution. As shown in the figure, all the distributions appear to be free-form and complex, showing the diversity of the augmentation policy searched by MCMC-Aug.