# OpenReview forum: "A Continuous Mapping For Augmentation Design"
_NeurIPS.cc/2021/Conference — NeurIPS 2021 Poster_

### Official Review · Reviewer_Dpxh · 2021-07-08

**Rating:** 7
**Confidence:** 4

**Summary:**

The paper presents an automated data augmentation technique that optimizes a distribution over a continuous augmentation space. Augmentation is formulated as learning a continuous mapping from an augmentation vector in R^d to an image transformation that composes three general families/modes of image transformations (color adjustment, image filtering, and image warping).   The proposed method uses a bilevel optimization approach where the primary learning task is trained on an augmented training set and the augmentation learning task is trained on the validation set. To make the optimization tractable, the inner level training is replaced by a one-step gradient update on an augmented mini-batch with model parameters being shared across the bilevel optimization iterations, i.e., incrementally updated during the bilevel optimization. A comprehensive set of experiments, including ablation experiments and comparisons to SOTA methods for automated data augmentation, demonstrate the efficacy and efficiency of the proposed method.

**Limitations And Societal Impact:**

The authors did not acknowledge any potential negative societal impact of their work.

**Main Review:**

The paper presents an efficient method for automated data augmentation that is learned end-to-end with the primary learning task without compromising performance. The paper moves away from restrictive assumptions about the discrete augmentation space, including predefined/handcrafted augmentations, and poses automated augmentation as a continuous optimization problem. This enables diversity in the augmented samples and avoids ad-hoc approximations when computing gradients in the discrete space cases. Experiments focus on natural image classification tasks, but the method is general to be adapted to other learning tasks and image modalities.

The work can be improved by addressing the below concerns:

- The method relies on the predefined semantic groupings of augmentation modes (color adjustment, image filtering, and image warping) that assume natural images, but the method is generic enough to be applied to other image modalities by considering modality-specific parameterized transformations. This should be made clear in the discussion section.

-  It would be insightful if a discussion is included on the choice of MCMC versus variational inference given the Bayesian formulation of the problem.

- The learned augmentation relies heavily on the generalization of the model on the validation set. There is no machinery in the proposed method that guarantees the generation of challenging augmented samples for the classification task (e.g., samples that can be easily misclassified, near the decision boundary).

- How sensitive/robust the proposed method under data shift and outliers?

- How sensitive/robust the proposed method under a severely limited training budget (e.g., medical imaging)? For instance, in Algorithm 1, a single augmented sample is added to the training data at each bilevel iteration, this won't be applicable in high-dimension-low-sample-size scenarios.








**Time Spent Reviewing:**

2.5

---

> ### Author Response · Authors · 2021-08-10
> **1st Response to Reviewer Dpxh**
>
> We are sincerely grateful to reviewer Dpxh for the valuable suggestion. We will explain their concerns point by point.
>
> 1. **[Q]** "The method relies on the predefined semantic groupings of augmentation modes (color adjustment, image filtering, and image warping) that assume natural images, but the method is generic enough to be applied to other image modalities by considering modality-specific parameterized transformations. This should be made clear in the discussion section."
>
>     **[A]** The reviewer made a constructive observation that our method is generic enough to other image modalities, such as infrared/X-rays/ultrasound imaging. We will add them to the discussion section in the final version.
>
> 2. **[Q]** "It would be insightful if a discussion is included on the choice of MCMC versus variational inference given the Bayesian formulation of the problem."
>
>     **[A]** Thanks for the suggestion. Both Variational Inference (VI) and MCMC methods are able to obtain the posterior in the Bayesian setting. In general, MCMC methods require sampling to directly approximate the posterior, while VI achieves this by manually picking an approximating distribution (e.g., exponential family) and minimize the distance between the approximating distribution and true posterior. Since automated machine learning aims to minimize human involvement in the algorithm, we choose MCMC over VI to minimize the amount of manual design. We would add this discussion to the final version.
>
> 3. **[Q]** "The learned augmentation relies heavily on the generalization of the model on the validation set."
>
>     **[A]** The same as previous automated data augmentation (ADA) techniques [R1, R2, R3, R4, R5], our method also relies on the generalization of the model on the validation set. Despite the concern of the reviewer, the validation set is still widely used by existing ADA works and ours because of its substantial benefits:
>
>     1. The gap between validation accuracy and test accuracy is small on datasets like ImageNet, CIFARs, and could be further reduced by increasing the number of examples in the validation set.
>     2. Using the validation accuracy as the learning objective removes the necessity of carefully selected metrics or designed pretext/proxy tasks to model the generalization. Modeling the generalization property of deep models accurately is still one of the most important unsolved problems in the area.
> 4. **[Q]** "There is no machinery in the proposed method that guarantees the generation of challenging augmented samples for the classification task (e.g., samples that can be easily misclassified, near the decision boundary)."
>
>     **[A]** "generation of challenging augmented samples" can be achieved but may not be the optimal choice. Details:
>
>     1. "generation of challenging augmented samples" can be achieved with minor modification of our solution. Specifically, this can be achieved by changing the current optimization objective from "minimizing the validation loss" to "maximizing the training loss" (like an adversarial training setting). In this way, more difficult augmented samples are encouraged to be generated, such as samples near the decision boundary.
>     2. Challenging augmented samples may not always benefit the generalization of the model. For example, adding addictive adversarial perturbation to the training data is a widely adopted approach to find challenging augmented data. However, as reported in [R6], the training with adversarial perturbation would undermine the generalization of the model on clean data.
> 5. **[Q]** "How sensitive/robust the proposed method under data shift and outliers?"
>
>     **[A]** We agree that data shift and outliers are common issues in the real world. In this paper, we mainly focus on the methodology of ADA, so we follow the same experimental settings of prior works to ensure comparability and do not introduce a robust study on the data shift and outliers. Nonetheless, we think this is worth considering and solving, and if some ADA approach could effectively address the issue of data shift and outliers, it would be another valuable work.
>
> 6. **[Q]** "How sensitive/robust the proposed method under a severely limited training budget (e.g., medical imaging)? For instance, in Algorithm 1, a single augmented sample is added to the training data at each bilevel iteration, this won't be applicable in high-dimension-low-sample-size scenarios."
>
>     **[A]** Deep learning (DL) applications in high-dimension-low-sample-size (HDLSS) scenarios are known to be a huge challenge since DL usually depends heavily on the quality and amount of data. This calls for other powerful data analysis techniques like feature selection/data dimensionality reduction [R7, R8] and learning methods like few-shot learning [R9]. We believe it is a promising direction to adapt the existing ADA to these areas.
>
> ---
>
> [R1] Cubuk, Ekin D., et al. "Autoaugment: Learning augmentation strategies from data." Proceedings of the IEEE/CVF Conference on Computer Vision and Pattern Recognition. 2019.
>
> [R2] Lim, Sungbin, et al. "Fast autoaugment." Advances in Neural Information Processing Systems 32 (2019): 6665-6675.
>
> [R3] Ho, Daniel, et al. "Population based augmentation: Efficient learning of augmentation policy schedules." International Conference on Machine Learning. PMLR, 2019.
>
> [R4] Lin, Chen, et al. "Online hyper-parameter learning for auto-augmentation strategy." Proceedings of the IEEE/CVF International Conference on Computer Vision. 2019.
>
> [R5] Tian, Keyu, et al. "Improving Auto-Augment via Augmentation-Wise Weight Sharing." Advances in Neural Information Processing Systems 33 (2020).
>
> [R6] Madry, Aleksander, et al. "Towards deep learning models resistant to adversarial attacks." *arXiv preprint arXiv:1706.06083* (2017).
>
> [R7] Chowdhury, Shanta, Xishuang Dong, and Xiangfang Li. "Recurrent neural network based feature selection for high dimensional and low sample size micro-array data." 2019 IEEE International Conference on Big Data (Big Data). IEEE, 2019.
>
> [R8] Mahmud, Mohammad Sultan, and Xianghua Fu. "Unsupervised classification of high-dimension and low-sample data with variational autoencoder based dimensionality reduction." 2019 IEEE 4th International Conference on Advanced Robotics and Mechatronics (ICARM). IEEE, 2019.
>
> [R9] Wang, Yaqing, et al. "Generalizing from a few examples: A survey on few-shot learning." ACM Computing Surveys (CSUR) 53.3 (2020): 1-34.

---

> > ### Comment · Reviewer_Dpxh · 2021-08-13
> > **Increasing my rating**
> >
> > The authors have addressed my concerns and provided compelling arguments and clarification. I hence increase my rating to "7: Good paper, accept."

---

> > > ### Author Response · Authors · 2021-08-13
> > > **Appreciation**
> > >
> > > Thank you for appreciating our efforts.
> > > We are glad that our response addressed your concerns and resolved the questions.
> > >
> > > A minor note — are you not supposed to edit your old score and update it to 7 because it still shows 6?

---

### Official Review · Reviewer_YW9j · 2021-07-16

**Rating:** 8
**Confidence:** 4

**Summary:**

    Authors present a continuous formulation of data augmentation for
    images. The formulation is used to define a Bayesian model for augmentation
    parameters. The Bayesian model does not rely on a bi-level optimization
    problem as commonly used for optimizing data augmentation. Instead, only a
    single gradient step approximates a full optimization of a network with a
    given set of augmentation parameters. Gradient step is taken starting from
    pretrained weights determined through initial training using random
    augmentation. Samples from the posterior distribution are gathered using
    Langevin dynamics MCMC with random batches. Samples are used to train
    the ultimate networks. Experiments show that the proposed model outperforms
    existing techniques in terms of accuracy of the final models at a fraction
    of computation time.

**Limitations And Societal Impact:**

    I do not see a negative societal impact of this work. On the contrary, this
    work may lead to reduction in training times and save energy.

    Authors claim they discuss the limitations of the work in Section 6 but I
    fail to see that in the text. The main limitation is the scope of the
    experiments. Authors could have demonstrated the model on larger set of
    problems, e.g., segmentation. They could have also used larger
    datasets. However, I do not see them as crucial limitations.



**Main Review:**

    The proposed technique is novel to the best of my knowledge. The continuous
    formulation, which is emphasized a bit too much perhaps in the article, is
    necessary for using Langevin dynamics MCMC during inference. The article
    deviates quite a bit from common literature on this topic and comes up with
    a novel method that yields great results at the fraction of the compute
    time.

    Experimental results are quite convincing. Authors only evaluate using
    rather small datasets but I believe this is the common strategy taken by
    most of the works in this domain, therefore, it is not a major drawback.

    The article is mostly well written. The method and experiments are clearly
    explained. The presentation can be improved but I do not think this is a
    major drawback.

    Data augmentation is an essential part for training neural networks. A
    computationally light method for determining best augmentation parameters
    would be very valuable in practice.

    My questions are mostly about the experimental setup:
    1. Authors show in Table 1 that the random sampling of the augmentation
       parameters already yields better results than some of the methods that
       optimize for the parameters. This is rather unexpected and
       not intuitive. Can they provide an explanation for this?
    2. It is unclear to me whether authors start all the augmentation methods
       from some pretrained model. Using the pretrained model is essential, as I
       understand it, for the approximation given in Equation 9. Without a
       pretrained model, the proposed approach would not work most
       probably. Such pretraining may also help the other methods for
       determining optimal augmentation parameters, both in terms of final
       accuracy and computation time. Have authors started the other methods
       from a pretrained model?
    3. It is somewhat surprising that a single gradient step approximates a full
       optimization so well in this work. Can authors elaborate on this?


      --- After the update ---
    I thank the authors for the responses. I keep my opinion that this is good work.

**Time Spent Reviewing:**

2.5; 3.5 with the rebuttal.

---

> ### Author Response · Authors · 2021-08-10
> **1st Response to Reviewer YW9j**
>
> We thank reviewer YW9j for their careful readings and valuable comments on our paper. We are grateful that the reviewer found value in our work. We'll address the questions about experiments or limitations as follows:
>
> 1. **[Q]** "Authors show in Table 1 that the random sampling of the augmentation parameters already yields better results than some of the methods that optimize for the parameters. This is rather unexpected and not intuitive. Can they provide an explanation for this?"
>
>     **[A]** The random augmentation policies defined in our continuous search space are very different from the augmentation policies in prior works that are in discrete space. Generally, we consider that our continuous search space has a stronger regularizing effect than selecting discrete augmentation magnitudes from a discrete space as the discrete augmentations would only be able to produce transformed data with specific magnitudes. Therefore, even the random continuous policies could outperform some optimized discrete policies. The results in Table 1 also show the benefit of using continuous augmentation policies in our work compared with the discrete augmentation policies in prior works.
>
> 2. **[Q]** "It is unclear to me whether authors start all the augmentation methods from some pretrained model. Using the pretrained model is essential, as I understand it, for the approximation given in Equation 9. Without a pretrained model, the proposed approach would not work most probably. Such pretraining may also help the other methods for determining optimal augmentation parameters, both in terms of final accuracy and computation time. Have authors started the other methods from a pretrained model?"
>
>     **[A]** The "pretrain-finetune" paradigm is firstly introduced by AWS [R3], which we also discussed and compared in our paper. We have discussed with authors of [R3], they confirmed that the "pretrain-finetune" paradigm can only be applied to the nested-optimization-based methods [R1, R2], and is not applicable to the others. It evaluates the augmentation policy by finetuning with some "specially pretrained weight" instead of training a newly initialized model. The "specially pretrained weight", which is trained with uniform augmentation distribution, is capable of approximating the training signal of hyperparameters which can only be computed by differentiating through a complex optimization procedure. We will add more descriptions on the "pretrain-finetune" paradigm in the final version.
>
> 3. **[Q]** "It is somewhat surprising that a single gradient step approximates a full optimization so well in this work. Can authors elaborate on this?"
>
>     **[A]** Using a single gradient step to approximate a full optimization has been investigated in meta-learning literature [R4, R5]. Besides, the two factors below make this approximation more reasonable:
>
>     1. Deep model's parameters are known to be more fixed in the later stage of the training process.
>     2. On the contrary, the augmentations in the later training stages are demonstrated to be more influential and important than earlier ones in [R3].
>
>     We will include a more detailed discussion in the final version.
>
> 4. **[Q]** "The main limitation is the scope of the experiments. Authors could have demonstrated the model on a larger set of problems, e.g., segmentation. They could have also used larger datasets."
>
>     **[A]** Thanks so much for the constructive advice. Besides the future work listed in Section 6. towards more diverse augmentations and wider applicants, e.g., in other computer vision areas or the times-series processing, problems on larger datasets are more challenging but rewarding. Since our method is demonstrated to be efficient and effective on the large-scale ImageNet dataset, we are confident that we could extend the success to larger real-world datasets. This is another important and valuable direction for our future work.
>
> ---
>
> [R1] Cubuk, Ekin D., et al. "Autoaugment: Learning augmentation strategies from data." Proceedings of the IEEE/CVF Conference on Computer Vision and Pattern Recognition. 2019.
>
> [R2] Cubuk, Ekin D., et al. "Randaugment: Practical automated data augmentation with a reduced search space." Proceedings of the IEEE/CVF Conference on Computer Vision and Pattern Recognition Workshops. 2020.
>
> [R3] Tian, Keyu, et al. "Improving Auto-Augment via Augmentation-Wise Weight Sharing." Advances in Neural Information Processing Systems 33 (2020).
>
> [R4] Contardo, Gabriella, Ludovic Denoyer, and Thierry Artières. "A meta-learning approach to one-step active learning." arXiv preprint arXiv:1706.08334 (2017).
>
> [R5] Finn, Chelsea, et al. "Online meta-learning." International Conference on Machine Learning. PMLR, 2019.

---

> ### Author Response · Authors · 2021-09-01
> **Gentle Reminder**
>
> Thanks again for your detailed review. This is a gentle reminder that the deadline for preliminary reviewing process is approaching soon. We are looking forward to your soonest reply.

---

### Official Review · Reviewer_wJZx · 2021-07-19

**Rating:** 7
**Confidence:** 4

**Summary:**

The authors pose the problem of learning data augmentation as a continuous optimization problem and introduces a Bayesian approach for learning and sampling augmentations. The method allows for optimizing over a space of infinitely many possible transformations in contrast to the majority of automated data-augmentation approaches (ADA) which are limited by discretization of the solution space. The values of the proposed approach over prior works in terms of prediction accuracy and speed have been demonstrated on a range of benchmarks.

**Ethics Review Area:**

["I don’t know"]

**Limitations And Societal Impact:**

The main limitations in my view are the lack of insights into the limitations of the proposed approach as elaborated above.
NA for societal impact.

**Main Review:**

Strengths
- Clarity: The manuscript is very well-written, and its contributions are clearly communicated with respect to prior works.
- Convincing empirical results:  the method has been compared against a range of relevant prior methods on multiple image classification benchmarks and demonstrates convincingly that it can attain a better accuracy-speed trade-off.
- Solid method contribution:  the proposed method is carefully constructed by mixing different ingredients, all of which are important in different ways. Firstly, differentiable parameterization of transformations is a necessary first step to pose the ADD as a continuous optimization problem. Secondly, the single-step approximation used in eq. (9) offers a computationally efficient approximation to the bi-level optimization problem of ADD. Lastly, the use of Stochastic Gradient Langevin Dynamics allows ones to approximately sampling multiple data-augmentation strategies from the posterior distribution instead of resorting to a point-estimate of the optimal data-augmentation strategy, a key to ensuring diversity of relevant transformations.


Weaknesses
- There is little insight into when the proposed method may break or may not work as well as other competing methods. How does the efficacy of the proposed method vary with the size of the validation dataset? When would the single-step approximation be assumed in eq. (9). Is there any relevant form of data augmentation that may be hard to parametrize in a differentiable way? If so and if such transformations were to be critical for some applications, then wouldn't the competing approaches based on discrete optimization be more preferable?
- It is unclear what the benefits of the Bayesian approach are. For example, one could set the noise term $\eta=0$ in eq.(13) and I suspect that the method may still work reasonably well (not shown in the manuscript). The noise injection may potentially improve the diversity of sampled augmentations, but I feel this should be shown to substantiate the benefits of the Bayesian approach.


Minor comments
- line 103: "differential" => "differentiable"
- line 124: "we pick k<< n for simplicity" -> what do we lose from this assumption?
- line 182: "pseudo one-step update" -> why pseudo?
- line 204: "second-order derivative … computed more efficiently" => efficient in time or space?
- line 232: no space after '.'
- line 255: "Tab. 3 and Tab. 2" => Tab. 2 and Tab. 3

**Time Spent Reviewing:**

6

---

> ### Author Response · Authors · 2021-08-10
> **1st Response to Reviewer wJZx**
>
> We would like to greatly appreciate reviewer wJZx's detailed and pertinent reviews. To their questions and concerns:
>
> 1. **[Q]** "There is little insight into when the proposed method may break or may not work as well as other competing methods. How does the efficacy of the proposed method vary with the size of the validation dataset? When would the single-step approximation be assumed in eq. (9)."
>
>     **[A]** To show how different validation sizes affect the method, we conduct the experiment by varying the size of the validation set. We changed the validation set size of CIFAR-10 from 10,000 to 256 and kept the rest of the experiment to be identical. As a result, a performance drop of around 0.4 (2.96 -> 3.31) is confirmed. We conclude that a too-small validation set may have poor generality, thus, is not reliable enough to be used for automatically searching for augmentations. However, being unable to conduct a good evaluation of the augmentation policy when validation data is lacking is a common challenge to current state-of-the-art methods, and it is worth further studying. We believe that if this is done, it would be another nice and solid work. We will add some discussions in the limitation/conclusion section.
>
> 2. **[Q]** "Is there any relevant form of data augmentation that may be hard to parametrize in a differentiable way? If so and if such transformations were to be critical for some applications, then wouldn't the competing approaches based on discrete optimization be more preferable."
>
>     **[A]** Data augmentations that are hard to parameterize are a clear limitation of our method. However, an approximation could be made to bypass this issue. The most straightforward approach would be applying linear interpolation between the original image and the augmented image. Suppose we have an image $I$ and a transformation $T$ (which is hard to parameterize), we introduce a continuous parameter $\lambda$ and create a new augmentation via $T^\prime(I) = \lambda\ T(I) + (1-\lambda)\ I$. We expect to see a better approach to resolve the limitation of augmentation parameterization in the future. It is a valuable direction and ought to be further explored.
>
> 3. **[Q]** "It is unclear what the benefits of the Bayesian approach are. For example, one could set the noise term $\eta=0$ in eq.(13) and I suspect that the method may still work reasonably well (not shown in the manuscript). The noise injection may potentially improve the diversity of sampled augmentations, but I feel this should be shown to substantiate the benefits of the Bayesian approach."
>
>     **[A]** We did find that the Bayesian approach is important in the problem under investigation. Theoretically, the SGLD would degenerate to SGD when the noise component is set to zero. Empirically, we set $\eta$ to zero on CIFAR-10 for WRN-40-2 and observe that the test error increased from 2.96 to 3.24. The obvious increase (around 10% relative increase) of error shows that the noise is essential to our method.
>
> 4. **[Q]** minor comments on typos
>
>     **[A]** Thanks for your kind comments. We have carefully checked and revised the whole paper several times according to your suggestion. All typos we found have been corrected.

---

> ### Author Response · Authors · 2021-09-01
> **Gentle Reminder**
>
> Thanks again for your detailed review. This is a gentle reminder that the deadline for preliminary reviewing process is approaching soon. We are looking forward to your soonest reply.

---

### Official Review · Reviewer_7Dyr · 2021-07-21

**Rating:** 4
**Confidence:** 3

**Summary:**

This contribution is aimed at automated data augmentation. A continuous search space is defined using simple image operators, and thus enables learning of augmentation parameters. A randomised algorithm is proposed for parameter learning.

**Limitations And Societal Impact:**

See above - no immediate adverse societal impact

**Main Review:**

Data augmentation is an important topic and automated learning of augmentation methods and parameters can improve deep learning.
In this paper the authors engineer a randomized algorithm for learning to augment.

The  originality could be better; there a numerous examples of continuous parameterizations of image augmentation starting with the work of Hauberg et al.  2016: "Dreaming more data: Class-dependent distributions over diffeomorphisms for learned data augmentation." In Artificial Intelligence and Statistics (pp. 342-350). PMLR, and later papers citing this one. Also the literature on generative modeling (GAN etc) is missing, see e.g. Antoniou et al 2017. Data augmentation generative adversarial networks. arXiv preprint arXiv:1711.04340.

The quality of the work is good, has a fine motivation and good experimental evaluation. Promising results. However, I recommend in a revised version to do a much more careful literature search and place the work more specifically in the literature on learned augmentations.

The clarity of the report is excellent, it is well written.

Overall the significance is limited by the novelty and also, if I may recommend to focus a bit more on analysis, to understand to the limitations of learning data augmentation.

**Time Spent Reviewing:**

3

---

> ### Author Response · Authors · 2021-08-10
> **1st Response to Reviewer 7Dyr**
>
> We sincerely thank reviewer 7Dyr for the comments. We appreciate that the reviewer found our work to be of high technical quality with good experimental results and excellent clarity. The concerns are mainly about the novelty.
>
> Regarding novelty, firstly we would like to highlight that the scope and the primary novelty (as also appreciated by the other reviewers) of our approach are as follows:
>
> 1. **[the scope]** Following the representative works [R1, R2], the scope of data augmentation we discuss refers only to the basic data augmentations (e.g., rotation, brightening, blurring, etc., and their combinations), which are the most commonly used augmentations within state-of-the-art works [R9, R10]. The works mentioned by the reviewer [R3, R4] propose advanced data augmentations like optical-flow-like [R3] or GAN-based [R4], which are out of the scope of basic augmentations. We will emphasize that the data augmentation mentioned in this paper only refers to the basic augmentations to improve the clarity.
>
> 2. **[novelty 1]** The advanced data augmentations [R3, R4] are continuously parameterized by design. However, these works cannot make all the basic augmentations continuous. We are the first to pose a continuous optimization problem for automatically learning these basic augmentations. We achieve this goal by introducing a generic parameterization method for a wide range of augmentations, and this method is totally different from [R3, R4]'s. As discussed in the paper, this could be regarded as a milestone for the development of automated data augmentation (ADA): most of the previous ADA methods are based on the discrete space of basic augmentations; some [R8] tried to make them continuous but introduced approximations like Gumbel-softmax, thus is still not able to explore the continuous space of basic augmentations. More differences between [R3, R4] and ours are detailed below.
>
> 3. **[novelty 2]** We propose Bayesian formulation and single-step approximation for ADA so that the highly efficient SG-MCMC is applicable. Specifically, the bilevel nature of augmentation learning hinders the application of the fast MCMC method in [R3]. In this paper, we are able to obtain a state-of-the-art augmentation policy efficiently even on large-scale dataset like ImageNet.
>
> The more detailed differences between [R3, R4] and ours are listed below:
>
> 1. Some existing works on the continuous parameterization of image augmentations [R3] **1)** focus on specifically proposed transformations and **2)** use proxy tasks like image alignment. Thus, they could hardly generalize to the rich variety of basic augmentations (such as rotation/blurring) since there exists intrinsic difference between these proposed transformations and basic augmentations. In contrast, our method could achieve this.
> 2. The mechanisms of GAN-based methods [R4, R5, R6, R7] and existing ADA methods (including ours) on how to generate augmented data are very different as well. As discussed by the first ADA work [R1], GANs directly generate the augmented data via neural networks, while ADA methods learn the hyperparameters of predefined basic augmentations and use them to augment the data (by rotation, blurring, etc.).
>
> Due to the space limitation, we could only manage to discuss the most closely related work (ADA and MCMC) in the original submission. Nevertheless, following the kind suggestion of the reviewer 7Dyr, we will include more references [R3, R4, R5, R6, R7] in the Related Work, to help place our work more properly.
>
> &nbsp;
>
> But as discussed above, these works would not devalue our work or other ADA methods which focus on the basic augmentations.
>
> &nbsp;
>
> ---
>
> [R1] Cubuk, Ekin D., et al. "Autoaugment: Learning augmentation strategies from data." Proceedings of the IEEE/CVF Conference on Computer Vision and Pattern Recognition. 2019.
>
> [R2] Cubuk, Ekin D., et al. "Randaugment: Practical automated data augmentation with a reduced search space." Proceedings of the IEEE/CVF Conference on Computer Vision and Pattern Recognition Workshops. 2020.
>
> [R3] Hauberg, Søren, et al. "Dreaming more data: Class-dependent distributions over diffeomorphisms for learned data augmentation." *Artificial Intelligence and Statistics*. PMLR, 2016.
>
> [R4] Antoniou, Antreas, Amos Storkey, and Harrison Edwards. "Data augmentation generative adversarial networks." *arXiv preprint arXiv:1711.04340* (2017).
>
> [R5] Perez, Luis, and Jason Wang. "The effectiveness of data augmentation in image classification using deep learning." *arXiv preprint arXiv:1712.04621* (2017).
>
> [R6] Mun, Seongkyu, et al. "Generative adversarial network based acoustic scene training set augmentation and selection using SVM hyper-plane." *Proc. DCASE* (2017): 93-97.
>
> [R7] Zhu, Xinyue, et al. "Data augmentation in emotion classification using generative adversarial networks." *arXiv preprint arXiv:1711.00648* (2017).
>
> [R8] Li, Yonggang, et al. "DADA: Differentiable automatic data augmentation." arXiv preprint arXiv:2003.03780 (2020).
>
> [R9] Tan, Mingxing, and Quoc Le. "Efficientnet: Rethinking model scaling for convolutional neural networks." International Conference on Machine Learning. PMLR, 2019.
>
> [R10] Brock, Andrew, et al. "High-performance large-scale image recognition without normalization." arXiv preprint arXiv:2102.06171 (2021).

---

> ### Author Response · Authors · 2021-09-01
> **Gentle Reminder**
>
> Thanks again for your detailed review. This is a gentle reminder that the deadline for preliminary reviewing process is approaching soon. We are looking forward to your soonest reply.

---

### Decision · Program_Chairs · 2021-09-27

**Decision:**

Accept (Poster)

**Comment:**

This paper is concerned with augmentation design, i.e. finding a set of parameters for generating random image variants via color adjustment / filtering / warping to aid generalization in models based on images. The basic idea is to reduce augmentations to a finite-dimensional continuous space, then learn a distribution over the parameters. State of the art methods [5] use bi-level optimization to learn this distribution. Instead, this paper proposes a Bayesian posterior where the likelihood of any set of hyperparameters can be computed in terms of scores on a held-out validation set. That construction, if defined exactly, would still involve a double-loop. However, essentially by "warm-starting" the inner-optimization, this is avoided and a single-loop sampling algorithm is obtained.

Reviewers were generally positive about this paper, and found it clear. The major concerns were that the experimental setup could be improved, discussion of related work could be more thorough, there could be more discussion of when the method might fail, and that it isn't entirely clear what the benefits of the Bayesian approach is. (The paper argues that it has the benefit of avoiding a double-loop optimization, but it appears that the idea of warm-starting the inner optimization could be applied to other methods as well, so it isn't entirely clear what is providing the benefit here.)

I found the central construction difficult to understand. (Reviews simply referred to it as a Bayesian model, but I wanted to understand what this model was, so I read the paper in detail.) After extensive discussion with the authors I am satisfied that there is a coherent model being defined. However, I strongly suggest that the notation of the paper be reconsidered to first explicitly state the model in the form of the equations (i), (ii), and (iii) that were given in the discussion below.